# Zinc Oxide Nanoparticles Can Intervene in Radiation-Induced Senescence and Eradicate Residual Tumor Cells

**DOI:** 10.3390/cancers13122989

**Published:** 2021-06-15

**Authors:** Nadine Wiesmann, Rita Gieringer, Melanie Viel, Jonas Eckrich, Wolfgang Tremel, Juergen Brieger

**Affiliations:** 1Department of Otorhinolaryngology, University Medical Center Mainz, Langenbeckstrasse 1, 55131 Mainz, Germany; gieringe@uni-mainz.de (R.G.); Jonas.Eckrich@ukbonn.de (J.E.); brieger@uni-mainz.de (J.B.); 2Department of Oral and Maxillofacial Surgery, Plastic Surgery, University Medical Center Mainz, Augustusplatz 2, 55131 Mainz, Germany; 3Department of Chemistry, Johannes Gutenberg-University, Duesbergweg 10-14, 55128 Mainz, Germany; viel@uni-mainz.de (M.V.); tremel@uni-mainz.de (W.T.)

**Keywords:** senescence, tumor therapy, radioresistance, zinc oxide nanoparticles

## Abstract

**Simple Summary:**

Despite great advancements in modern cancer therapy, many patients suffer from local recurrence after initial treatment. The phenomenon of regrowth of tumor cells after several months or years is increasingly connected to therapy-induced cellular senescence. Our goal was to investigate the properties of tumor cells after survival of 16 Gy gamma-irradiation. We revealed the classical hallmarks of senescence among the remnant cell mass after irradiation. Furthermore, the observed radiation-induced senescence was associated with the increased ability to withstand further irradiation and cells were shown to possess the ability to regrow within several weeks. Moreover, treatment with zinc oxide nanoparticles proved to be an attractive therapeutic option to eradicate the residual senescent tumor cells.

**Abstract:**

Despite recent advancements in tumor therapy, metastasis and tumor relapse remain major complications hindering the complete recovery of many cancer patients. Dormant tumor cells, which reside in the body, possess the ability to re-enter the cell cycle after therapy. This phenomenon has been attributed to therapy-induced senescence. We show that these cells could be targeted by the use of zinc oxide nanoparticles (ZnO NPs). In the present study, the properties of tumor cells after survival of 16 Gy gamma-irradiation were investigated in detail. Analysis of morphological features, proliferation, cell cycle distribution, and protein expression revealed classical hallmarks of senescent cells among the remnant cell mass after irradiation. The observed radiation-induced senescence was associated with the increased ability to withstand further irradiation. Additionally, tumor cells were able to re-enter the cell cycle and proliferate again after weeks. Treatment with ZnO NPs was evaluated as a therapeutical approach to target senescent cells. ZnO NPs were suitable to induce cell death in senescent, irradiation-resistant tumor cells. Our findings underline the pathophysiological relevance of remnant tumor cells that survived first-line radiotherapy. Additionally, we highlight the therapeutic potential of ZnO NPs for targeting senescent tumor cells.

## 1. Introduction

Cancer is currently the leading cause of premature death between the ages of 30 and 69 years in Canada, the US, Australia, and most countries in Europe, including France, Germany, and the United Kingdom [1]. Despite recent advancements in tumor therapy, metastasis and local recurrence are major hurdles to completely curing cancer. After initial treatment, a considerable proportion of cancer patients suffer from relapse years or even decades later. Local recurrence adversely affects both long-term survival and patient quality of life in several malignancies [2]. Local control of the tumor is essential to prevent recurrences and inhibit the migration of tumor cells leading to distant metastases [2]. Depending on the entity, between 50% and 90% of cancer deaths are caused by metastases, underscoring the importance of the complete local control of the primary tumor and detection of the cancer as early as possible [3].

The underlying biological mechanisms that allow dormant residual tumor cells to be reactivated remain largely elusive. However, it is evident that the dormant cells are halted in a quiescent state and thus cannot be targeted by classical anti-cancer genotoxic treatment options which are designed to kill rapidly proliferating cells [4]. In research, a connection between tumor dormancy and the phenomenon of cellular senescence is becoming increasingly clear [5].

At first, senescence was described as so-called replicative senescence [6]. However, successive replication is not the only stressor that can initiate a senescent state. Stress-induced premature senescence (SIPS) [7] may result from the accumulation of oxidative damage to different cellular components, including DNA damage. Recently, the concept of therapy-induced senescence (TIS) has been proposed [7]. Tumor therapies are limited by their unintended side effects on the healthy tissue. Thus, some tumor cells may suffer sub-lethal damage that can lead to the initiation of TIS. Both chemotherapy-induced [8] and radiation-induced senescence [9,10] are known. Originally, senescence was defined as a state of dormancy in which re-entry into the cell cycle and return to cellular proliferation is prohibited. Thus, at first glance, tumor cells entering senescence seemed to represent a favorable condition as senescent cells cease to proliferate. Unfortunately, therapy-induced senescence in tumor cells is not necessarily a permanent cell fate and tumor cells were shown to re-enter the cell cycle under certain circumstances [4,5,11,12,13,14]. Radiation-induced tumor shrinkage and dormancy of the tumor cells can produce a remnant tumor mass which persists for long periods of time. In vivo data support the idea that these are senescent cells [15,16]. They create a microenvironment which can (i) favor the propagation of senescence, (ii) favor the escape from senescence, and iii) enhance the proliferation and dissemination of escaped cancer cells [4]. Often the repopulation is associated with the regrowth of tumor cells with enhanced ability to evade current therapy strategies and/or with higher clonogenic growth compared to the original tumor cell population [17]. Thus, tumor cell senescence cannot be defined as a state of being cured, but it rather should lead to further therapies capable of eliminating all remaining dormant tumor cells.

In the last few decades, the medical applications of nanoparticles have made great progress and several nanoparticulate formulations have already found their way into clinical practice [18]. Besides their known antibacterial and antifungal features, zinc oxide nanoparticles (ZnO NPs) have been shown to have favorable properties for application as innovative anti-tumor agents [19,20,21]. Various in vitro studies have shown that ZnO NPs can selectively damage tumor cells to some extent [22,23,24,25,26,27,28]. ZnO NPs are attractive because of their good biocompatibility, which results from the fact that zinc is an important trace metal of the human body that can be easily excreted. Moreover, ZnO NPs can be synthesized inexpensively in different sizes and morphologies, and their properties can be tuned by different modifications and coatings [29].

The purpose of this study was to explore the induction of radiation-induced cellular senescence and its characteristics in two different tumor cell lines. Our study highlights the importance of therapeutic targeting of residual tumor cells that survived first-line radiotherapy, as they can re-enter the cell cycle and proliferate again. Furthermore, we analyzed the cytotoxicity of ZnO NPs exerted in these tumor cell lines after entry into therapy-induced senescence and the potential of ZnO NPs to attack both proliferating and dormant tumor cells.

## 2. Results

In order to investigate the effects of gamma-irradiation on the cellular level, in vitro cultures of the tumor cell lines A549 (non–small cell lung cancer) and HuH-7 (hepatocellular carcinoma) were treated with 16 Gy and studied with respect to cell death, cell cycle arrest, and proliferation as well as morphological characteristics. 

### 2.1. Gamma-Irradiation with 16 Gy Results in Cell Death and Cell Cycle Arrest of the Remnant Tumor Cells

About twenty to thirty percent of the cells of both tumor cell lines died within 72 h after a single 16 Gy gamma-irradiation treatment (Figure 1). The most prominent cell death was apoptosis, while necrosis occurred to a lesser extent in both cell lines. HuH-7 cells showed a lower number of living cells in the untreated control sample in comparison to A549 cells; however, the amount of cell death following gamma-irradiation was comparable in both cell lines. 

The tumor cells that were able to withstand 16 Gy gamma-irradiation experienced an arrest of their cell cycle in the G2 phase (Figure 2). Both tumor cell lines showed significantly increased cell numbers in the G2 phase within one day after irradiation. Simultaneously, the percentage of cells in the G1 phase was decreased. The relative proportion of cells in the G2 phase doubled from 31% to 59% in A549 cells. In HuH-7 cells, the effect was even more pronounced, with the percentage of cells in the G2 phase increasing from 30% to 63%. Furthermore, within six days after irradiation, we observed a significant increase in polyploid cell numbers. The proportion of polyploid cells increased from 11% in non-irradiated cells to 27% on day six after 16 Gy in A549 cells and from 19% to 48% in HuH-7 cells.

### 2.2. Remnant Tumor Cells after 16 Gy Gamma-Irradiation Show the Classic Hallmarks of Senescence 

In addition to the effects on the cell cycle, typical changes in cell morphology occurred after irradiation (Figure 3). The cells exhibited increased size, they seemed to be swelling, and their morphology was reminiscent of fried eggs, with a protruding nucleus surrounded by a flattened cytoplasm. 

The observed increase in cell size was reflected in an increase in the mean cell diameter of the tumor cells (Figure 4). The diameter of untreated A549 cells was 20.9 µm ± 6.5 µm on average, while the diameter of untreated HuH-7 cells was 23.2 µm ± 8.0 µm. From the day of irradiation, a steady increase in the cell diameter was observed in both cell lines until day seven. On day seven after irradiation, the average cell diameter of A549 cells reached 61.1 µm ± 23.2 µm, and the diameter of HuH-7 cells reached 66.3 µm ± 32.4 µm. Thus, the average increase in cell size after irradiation was 43.20 µm ± 2.5 µm in A549 cells and 43.1 µm ± 3.6 µm in HuH-7 cells. Furthermore, the spectrum of cell diameters became larger after irradiation. Additionally, microscopic observation also showed an increase in the number of polyploid cells after irradiation, which was also observed with the cell cycle distribution measurement. 

The observed morphological changes after gamma-irradiation were associated with the arrest of proliferation. Proliferation was quantified by the determining cell numbers on subsequent days in cell cultures of untreated cells and in cell cultures of cells that had undergone irradiation (Figure 5). Untreated A549 cells showed a doubling of the cell count every 24 h. Proliferation of untreated HuH-7 cells was slightly slower. Counts of irradiated A549 cells stagnated on the fourth day (4 d) after irradiation. The count of irradiated HuH-7 cells slightly decreased from day four to day six after irradiation. 

The cease in tumor cell proliferation was also associated with a decrease in Ki-67 expression in irradiated tumor cells, as analyzed by immunohistochemical staining (Figure 6). Loss of Ki-67 expression in addition to the arrest of proliferation is also considered a classic feature of cellular senescence [30]. Loss of Ki-67 expression after irradiation was strong (reduction from 95% to 15% positive cells) in A549 cells and less pronounced but still considerable in HuH-7 cells, which showed a halving of Ki-67 positive cells. 

To support the hypothesis that the cellular changes observed after irradiation were indicative for cellular senescence, a senescence-associated β-galactosidase staining was performed (Figure 7). Positive staining is considered as one of the classic indicators of cellular senescence. Indeed, the staining clearly showed that the cells that exhibited swelling after gamma-irradiation were positive and were characterized by the typical blue staining. After irradiation with 16 Gy, almost all A549 cells became X-Gal-positive (99%) and most HuH-7 showed positive staining (81%). The staining was more pronounced near the nucleus than in other regions of the cytoplasm. 

To analyze cellular signaling typically associated with senescence, the expression levels of p21, p53, and p16 were determined by Western blotting (Figure 8). Expression levels of all three proteins were detected over time and compared. The expression in non-irradiated cells (0 Gy) served as a baseline and was compared to the expression on the first day after irradiation (16 Gy, 1 d) and on day six after irradiation (16 Gy, 6 d). Neither A549 nor HuH-7 cells expressed p16, in the irradiated nor in the non-irradiated state. HeLa cells, which bear high levels of p16, served as a control and showed p16 expression (Figure 8C). 

In A549 cells, an increase in p53 and p21 expression after irradiation was observed. In contrast, HuH-7 cells exhibited an increase in p21 expression after irradiation, but in these cells, no increase in p53 expression was detectable. Conversely, also non-irradiated HuH-7 cells showed a very high expression level of p53 compared to A549 cells, which did not increase further after irradiation. HuH-7 cells have been described to bear a p53 mutation [31,32], which could play a role in interpreting these signaling pathways.

### 2.3. The Senescent Phenotype Is Associated with Radiation Resistance 

Next, we examined how A549 cells and HuH-7 cells that had received 16 Gy irradiation and survived the treatment responded to a second treatment with 16 Gy six days after the first irradiation cycle (Figure 9). We found that six days after the first irradiation treatment, the cell population of A549 cells contained about 80% viable cells, while the viability of the HuH-7 cell population was only about 55%. After a second treatment with 16 Gy, the percentage of living cells of both cell lines did not decrease further. This indicates that repeated irradiation of senescent cells is not sufficient to induce cell killing. 

### 2.4. Senescent Cells Can Escape Dormancy after Weeks and Repopulate 

The next step addressed the controversial question as to whether tumor cells, once they have entered cellular senescence, remain in their inactive, non-proliferating state or whether they can regrow after days, weeks, or months. To answer this question, tumor cells remained in their cell culture flask after irradiation with 16 Gy. The cell culture medium was changed once a week, and flasks were repeatedly visually examined for their cellular proliferation. During the first one to two weeks, no changes were observed on microscopic examination. Apart from some cells that went into cell death and disappeared, the remaining cell population showed the above-mentioned hallmarks of senescent cells and the corresponding typical phenotype. Within the second to forth week after irradiation, we first noticed small islands of tumor cells that lost the senescent phenotype. Those cells were characterized by rapid proliferation indicated by cell division, and subsequentially the islands grew. Senescence-associated β-galactosidase staining revealed that the cells in the islands were becoming negative, i.e., the cells left the senescent state. Representative images of the cell islands and the surrounding cells of cell cultures of A549 and HuH-7 cells several weeks after single 16 Gy gamma-irradiation are shown in Figure 10. 

To support our hypothesis that previously senescent cells were able to grow again, we performed a cell cycle analysis of untreated tumor cells and irradiated tumor cells, one and six days after irradiation, and after several weeks (Figure 11). Cell cycle analysis confirmed our expectation: immediately after irradiation, A549 cells and HuH-7 cells entered G2 cell cycle arrest (1 d). Senescence was indicated by the G2 arrest and an increase in cells with multiple nuclei. When we detected islands of regrowing cells, the cell cycle distribution reversed, i.e., cells re-entered the G1 phase (28 d). After regrowth of cells, the number of polyploid cells increased in both cell lines. Multi-nucleation and an increase in polyploid cells has also been associated with oncogene-induced cellular senescence [33].

In order to evaluate the multiplication potential of the regrown tumor cells, we measured the proliferation of tumor cells that survived 1 × 16 Gy and 2 × 16 Gy and compared it to the proliferation of the original tumor cell population before treatment (Figure 12). The doubling time of A549 tumor cells before 16 Gy gamma-irradiation was 21.8 h. After survival of 1 × 16 Gray gamma-irradiation and regrowth of new colonies of non-senescent cells the doubling time of the regrown cells was 24.7 h. Thus, the regrowing cells achieved proliferation rates which were comparable to the original population; however, the multiplication of the tumor cells slowed down slightly. The doubling time of A549 cells after 2 × 16 Gy was 31.6 h, and thus the multiplication potential of these cells was decreased in comparison to the original cell population. Futhermore, the onset of regowth was delayed with the number of irradiation cycles. After 1 × 16 Gy, colonies of proliferating tumor cells were seen as soon as nine days after irradiation, while after 2 × 16 Gy, colonies appeared 22 days after the first irradiation cycle (16 days after the second irradiation) at the earliest. 

### 2.5. ZnO NPs Can Induce Cell Death in Senescent Cell Populations

Knowing that A549 and HuH-7 cells can enter a senescent state following gamma-irradiation, making them insensitive to further irradiation treatment, and knowing that these senescent cells can regrow after several weeks and form new viable colonies, we searched for a strategy to target senescent tumor cells. 

Preliminary studies by our own working group already showed that ZnO NPs were able to induce cell death in tumor cells. Therefore, we assessed whether ZnO NPs also exerted cytotoxic effects on senescent tumor cells. The ZnO NPs used in this study had a size of 10.73 nm ± 0.26 nm, which was determined with TEM. Our treatment protocol consisted of a first irradiation with 16 Gy on day zero, followed by tracking of the induction of senescence. On day six after irradiation, cells were treated with 100 µg/mL ZnO NPs for four hours. 

Following treatment with ZnO NPs, the percentage of vital cells, dead cells, apoptotic cells, and necrotic cells was determined 24, 48, and 72 h after treatment by staining with PI and FITC-annexin (Figure 13). Cells without staining were considered vital and double-stained cells were considered dead. Single-staining with FITC-annexin was considered to indicate an apoptotic cell death while single-staining with PI was considered to indicate a necrotic cell death. Using flow cytometric analysis of the stained cell populations, we could show that ZnO NPs could induce pronounced cell death in senescent A549 and HuH-7 cells. 

ZnO NPs could significantly decrease the percentage of vital cells in the senescent cell population in both cell lines. Within one day after treatment (24 h), cell viability dropped from approximately 80% to 30% in A549 cells and from approximately 60% to 30% in HuH-7 cells. Concomitantly, we could show a significant increase in the percentage of dead cells in the cell population of HuH-7 and A549 cells. The percentage of dead cells in the cell population reached about 55% in A549 cells and 50% in HuH-7 cells within one day after treatment (24 h). In HuH-7 cells, the percentage of apoptotic cells increased significantly after treatment with ZnO NPs, while in A549 cells, quick and pronounced cell death made it difficult to distinguish whether apoptotic or necrotic cell death was the more predominant cell death mechanism. Additional simultaneous application of 16 Gy with the ZnO NP did not increase tumor cell death in any of the senescent tumor cell populations.

In summary, we were able to show that ZnO NPs could efficiently induce cell death in senescent cell populations of A549 and HuH-7 tumor cell lines, which were resistant to further treatment with irradiation. Thus, ZnO NPs proved to be a new option to intervene in radiation-induced senescence and to eradicate residual tumor cells, which gained radiation resistance.

## 3. Discussion

Here, we show in a cell culture model that remnant tumor cells that survived 16 Gy gamma-irradiation can be targeted and killed by means of the use of ZnO NPs. Tumor recurrence after therapy still represents one of the major obstacles in the treatment of cancer. Sporadically, tumor cells are able to withstand the treatment regimens and after a period of time dormant tumor cells can proliferate again. This phenomenon in a clinical setting is increasingly being linked to cellular senescence [5]. Originally, senescence was defined as a state of permanent cell cycle arrest in which a return to cellular proliferation is prohibited. However, recent studies indicated that this perspective has to be re-evaluated [5,11,12,13,14]. Taking into account these findings, our goal was to study the characteristics of tumor cells after the survival of 16 Gy gamma-irradiation and to evaluate the potential of ZnO NPs as a therapeutic agent for targeting the remnant tumor cells. 

After treatment of A549 and HuH-7 cells with 16 Gy, following the increasing acceptance of high single-dose irradiation in the clinic [34,35], we were able to detect the typical features of senescence in remnant tumor cells. These included a cease in proliferation, which was reflected by constant cell numbers in the cell cultures, G2 cell cycle arrest, and decreased Ki-67 staining. Furthermore, the tumor cells exhibited typical morphological changes which are known to be associated with senescence: an increase in cell size and a typical “fried-egg-morphology”. Additionally, senescence-associated ß-galactosidase staining revealed positive staining of the designated senescent cells, which reflects an increased lysosomal activity in the irradiated cell populations. As current senescence markers are rather nonspecific, the condition of cellular senescence was determined by an accumulation of different characteristics. Senescence is viewed as a complex cellular state which is characterized by heterogenous phenotypic properties [36]. To address this problem, it is preferable to validate the senescent state of the cells by evaluating different phenotypical and molecular parameters. [30]. More specifically, (1) the cell cycle arrest, (2) the morphological characteristics, (3) the increased lysosomal compartment activity, and (4) the cellular signaling represented by the analysis of the expression of p16, p21, and p53 were the characteristics utilized for the determination of cellular senescence. 

The initiation of premature radiation-induced cellular senescence has already been described previously in A549 cells [37], whereas in HuH-7, only chemotherapy-induced senescence has been investigated previously [38]. Our findings indicate that both cell lines can follow the pathway of therapy-induced cellular senescence. 

Our protein expression analysis of p53, p21, and p16 revealed that senescence was initiated in both cell lines under investigation via the p53–p21 axis without involving the p16 protein. The p16 protein is regarded as one of the classic biomarkers of cellular senescence [30,39]. The protein is a cyclin-dependent kinase inhibitor (CDKI) and as such inhibits the progression of the cell cycle by inhibition of the cyclin D1-associated kinases Cdk4 and Cdk6, which promote the progression from the G1 to the S phase [39]. However, p16 could not be detected in A549 cells nor in HuH-7 cells, neither in non-irradiated nor in irradiated cells, which is in line with previous findings: A549 cells bear a deletion of the INK4a/ARF locus [40] from which the p16 protein is transcribed and the corresponding promoter region is hypermethylated in HuH-7 cells [41]. The lack of p16 expression in both cell lines under investigation is in line with our finding, that instead of a G1 cell cycle arrest, an arrest in the G2 phase was found in A549 as well as HuH-7 cells. This further indicates that, in contrast to the classic G1 exit in senescent cells, cell cycle arrest was initiated differently in these cell lines. Thus, we analyzed the expression of p21 and p53 in both cell lines, as senescence can also be initiated via the p53–p21 axis, and p21 was found to be involved in the G2/M checkpoint regulation [39]. Indeed, we found that p21 was elevated in A549 cells as well as HuH-7 cells after irradiation and initiation of the senescence phenotype. The elevation of the p21 level in A549 cells was also accompanied by an increase in the p53 expression, which is in line with the classic role of the wt-p53 of A549 cells in the DNA damage response [42]. These findings are supported by other studies that also associated therapy-induced senescence in A549 cells with increased expression of p53 and p21 [37,43]. In contrast, HuH-7 cells bear a mutated p53 protein, which was associated with a high expression of the protein [31,32]. The fact that p21 was activated in HuH-7 cells after irradiation despite the known mutation in the p53 protein indicates the existence of p53-independent signaling pathways triggering an increase in the p21 protein level as suggested before [44]. In summary, we showed that cellular senescence can be initiated by different signaling pathways depending on the genetic endowment of the respective cells. 

One controversial question is whether senescence renders tumor cells insensitive to further treatment or even favors the regrowth of more aggressive tumor cells. Senescent cells are suspected to create a microenvironment which can (i) favor the propagation of senescence in adjacent tumor cells as well as healthy cells [45], (ii) favor the escape from senescence, and (iii) enhance the proliferation and dissemination of escaped cancer cells [4,10]. Tumor cell senescence has already been linked to epithelial to mesenchymal transition, ‘stemness’, and radioresistance [46,47]. Therefore, we investigated how senescent A549 and HuH-7 cells respond to a second cycle of gamma-irradiation. Indeed, we found that a second treatment with 16 Gy gamma-irradiation failed to decrease the percentage of living cells in the cell populations that had survived the first cycle of irradiation. This is a strong indication for the acquisition of radioresistant properties. One possible explanation for this finding might be that the non-proliferative senescent cells are less sensitive to irradiation, as it is known that non-proliferating cells are less susceptible to irradiation due to their altered metabolism. Furthermore, the DNA of senescent cells might be better protected against radiation-induced DNA damage because the cells are arrested in the cell cycle [48]. It has been previously described that tumor cells show more aggressive properties after escape from senescence [14]; however, information on the doubling times and proliferation behavior of tumor cells in vitro after escape from therapy-induced senescence is sparse [17]. Our study shows that A549 cells can proliferate again following regrowth after 1 × 16 Gy and 2 × 16 Gy, but with lower growth rates compared to the original cell population. However, proliferative capacity alone cannot be taken as the sole measure of cell aggressiveness, as factors such as migration and invasiveness as well as the release of different factors may also be relevant.

Previous studies by our research group already showed that ZnO NPs can exert cytotoxic effects on tumor cells [22,49]. There is evidence that ZnO NPs in certain concentrations can damage tumor cells more than the surrounding cells of the normal tissue [22,23,24,25,26,27,28]. In addition, there is the possibility of surrounding the particles with a protective silica shell, which delays the toxicity of the particles. With the help of targeting moieties, preferred localization to the tumor tissue can be achieved and ZnO NPs can there exert their toxic potential. To follow up on these results, we treated the senescent A549 and HuH-7 cells with ZnO NPs and measured the percentage of apoptotic, necrotic, dead, and living cells in the cell population. We found that, in contrast to a second round of 16 Gy gamma-irradiation, ZnO NPs were able to decrease the percentage of vital cells in the cell population significantly and increase the percentage of dead cells within 24 h after treatment. In HuH-7 cells apoptotic cell death was more prominent, while in A549 cells the cell death mechanism was difficult to determine. This revealed that ZnO NPs can target senescent tumor cells. Thus, it can be assumed that their mechanism of action is not dependent on the availability of highly proliferative tumor cells. This is consistent with previous findings indicating that the doubling time alone is not sufficient to explain the sensitivity of tumor cells to treatment with ZnO NPs [25].

To our knowledge, this is the first study evaluating the cytotoxic effects of ZnO NPs on senescent tumor cells. The above-mentioned findings make ZnO NPs a very interesting drug to intervene in radiation-induced cellular senescence and to supplement modern radiotherapy. The combined treatment of tumor cells with ZnO NPs and irradiation might reduce the number of cells that can resist treatment and enter a dormant state. Additionally, our study underlines the importance of paying more attention to dormant tumor cells that have survived the first-line treatment.

## 4. Materials and Methods

### 4.1. Tumor Cell Lines and Culture Conditions

The non-small cell lung cancer (NSCLC) cell line A549 were purchased from DSMZ (German Collection of Microorganisms and Cell Cultures, Braunschweig, Germany). The hepatocellular carcinoma cell line HuH-7 has been originally purchased from the RIKEN BioResource Center. Cells were maintained in DMEM/Ham’s F12 (Sigma-Aldrich, St. Louis, MO, USA) supplemented with 5% BCS (VWR seradigm bovine calf serum iron-supplemented 100% US Origin, Cat. No. 10158-358, VWR^®^ Life Science, Radnor, PA, USA) and antibiotics (100 U/mL penicillin and 100 mg/mL streptomycin, Sigma-Aldrich, St. Louis, MO, USA) at 37 °C in 5% CO_2_.

### 4.2. Irradiation and Induction of Senescence

For irradiation, tumor cell lines were isolated by tryptic digestion (Sigma-Aldrich, St. Louis, MO, USA), and 700,000 cells were seeded in culture dishes with a growth area of 25 cm^2^. After 24 h, cells were irradiated with 16 Gy using a Cs137 gamma source. Dosages of 10 Gy gamma-irradiation are typically used to induce irradiation-induced senescence [50], which were shown to induce just under 80% senescent cells in A549 cell line [37]. As stereotactic body radiation therapy (SBRT) is gaining increasing acceptance in clinical practice, in which a single or up to five high dosages are applied [51,52], we chose a treatment with 16 Gy for our experiments. Preliminary studies of our research group showed that this dosage was high enough to recover a large percentage of senescent cells without losing too much of the cell population. A549 cells were cultivated for another 48 h in the same culture dish. Then, they were isolated by tryptic digestion and 250,000 cells were seeded in culture dishes with a growth area of 25 cm^2^. HuH-7 cells remained in the same culture dish due to pronounced cell death after irradiation. On day 6 after irradiation, the senescent state of the cells was checked prior to further interventions (nanoparticle treatment and/or second irradiation cycle).

### 4.3. Nanoparticle Synthesis

The solvothermal synthesis of zinc oxide nanoparticles (ZnO NPs) was adapted from Cheng et al. with some modifications [53]. In this process, 5 mmol of Zn(Ac)_2_·2H_2_O was dissolved in 10 mL of methanol by sonication, 20 mL of tetramethylammonium hydroxide 25% (*w/w*) in methanol was added slowly, and the mixture was stirred for 20 min. The reaction mixture was transferred to a 50 mL Teflon-lined stainless-steel autoclave and heated at 50 °C for 24 h. The colorless precipitate was separated by centrifugation and washed twice with deionized water and finally with ethanol. Finally, the product was dried in air, and stored in aliquots at 8 °C for storage.

### 4.4. Nanoparticle Characterization

The ZnO NPs were characterized by X-ray diffraction and transmission electron microscopy (TEM). X-ray diffraction patterns were recorded on a STOE Stadi P (Germany) diffractometer equipped with a Mythen 1k detector using MoKα_1_ radiation. Crystalline phases were identified using Match!3 software. Samples for TEM were prepared by placing a drop of NPs dispersion in ethanol on a carbon coated copper grid. TEM images for the characterization of size and morphology were obtained using a FEI Tecnai 12 microscope equipped with LaB_6_ source at 120 kV and a twin-objective together with a Gatan US1000 CCD-camera (2k × 2k pixels). Details of the nanoparticle characterization were previously published [22].

### 4.5. Treatment of Tumor Cell Lines with Nanoparticles

10 mg/mL ZnO NPs dispersions were freshly prepared immediately before each experiment with high-purity water (Aqua ad iniectabilia, B. Braun Melsungen AG, Melsungen, Germany). To disperse the NP, they were sonicated 5 min with 220–240 V and 37 kHz in an Elmasonic S 40 sonicator (Elma Schmidbauer GmbH, Singen, Germany). ZnCl_2_ solution was obtained from Sigma-Aldrich (0.1 M solution, Sigma-Aldrich, St. Louis, MO, USA) and diluted to the proper concentration with high-purity water. Cells were treated with the indicated amount of ZnCl_2_ or ZnO NPs for 4 h, followed by an exchange of the cell culture medium. Water served as control.

### 4.6. Senescence-Associated ß-Galactosidase Staining

Senescence-associated ß-galactosidase staining was performed with the Senescence ß-Galactosidase Staining Kit (#9860 Cell Signaling Technology^®^, Danvers, MA, USA) according to manufacturer’s instructions. It is based on the chromogenic substrate X-Gal (5-bromo-4-chloro-3-indolyl-βD-galactopyranoside). Upon exposure to X-Gal, a lysosomal hydrolase which is active in senescent cells at pH 6.0 produces a blue precipitate, which can be detected by light microscopy. In short, the cell culture medium was discarded, cells were washed with PBS, and then fixed with fixative solution. Prior to staining, cells were washed again with PBS and stained with the staining solution, taking care to maintain the pH at exactly 6. Cells were incubated overnight at 37 °C. For long-term storage, the staining solution was removed, cells were overlaid with 70% glycerol, and stored at 4 °C.

### 4.7. Light Microscopy

For light microscopic imaging of cellular morphology, proliferation, and cell numbers after irradiation and NPs treatment, cells were seeded in 25 cm^2^ cell culture flasks and treated with NPs according to the standard procedure. Images were taken prior to the treatment and at the indicated timepoints after treatment with a fluorescence microscope AxioVert 200M (Zeiss, Jena, Germany). Cells treated with an equivalent amount of water served as the control.

### 4.8. Measurement of Proliferation and Doubling Times

To harvest proliferating tumor cell colonies within the senescent cell populations, we waited until the colonies were visible with the naked eye (approximately 50–200 cells per colony). Then, they were picked with the help of a pipette tip, separated, and propagated. For measurement of the proliferative capacity of the tumor cells before and after 1 × 16 Gy and 2 × 16 Gy gamma-irradiation, 2000 cells/cm^2^ were seeded in 6-well-plates or 25 cm^2^ cell culture flasks. Beginning after 24 h, for four consecutive days, several pictures of the cells were taken at random areas of cell culture plates and the number of cells per area was determined. The average cell number/area was then calculated. Using the software GraphPad Prism 6 for Windows, Version 6.01 (GraphPad Software, La Jolla, CA, USA), the cell counts were used to extrapolate the doubling time of the cells using the exponential growth equation Y = Y_0_ * e^(K * x). K is the rate constant, expressed in reciprocal of the x-axis time units, and the doubling-time was computed as ln(2)/K.

### 4.9. Immunohistochemical Staining

Immunohistochemical analysis of tumor cell lines was performed according to standard procedures. In brief, 250,000 cells were seeded on coverslips in small Petri dishes, handled according to treatment procedures, and fixed in 4% paraformaldehyde/PBS, followed by endogenous peroxidase blockage with 3% H_2_O_2_/methanol. After preincubation with 10% normal serum in 2% albumin bovine/PBS for 20 min to avoid unspecific binding, the monoclonal primary mouse antibody anti-human Ki-67 Antigen Clone MIB-1(Dako Agilent, Santa Clara, CA, USA) was incubated overnight at 4 °C in a 1:150 dilution, followed by a biotinylated secondary antibody (Dako Agilent, Santa Clara, CA, USA), diluted 1:250 in streptavidin horseradish peroxidase. All washing procedures were performed in PBS. Slides were counterstained by hematoxylin. Cells incubated without the primary antibody served as negative controls. In the slides, five regions of interest (ROIs) were randomly chosen. The immunohistochemical staining was evaluated by two different observers independently (J.E. and N.W.), including at least 150 cells in each treatment group.

### 4.10. Flow Cytometry

For flow cytometric analysis cells were seeded in culture dishes with a growth area of 25 cm^2^. After treatment with the indicated amount of irradiation or NPs, cells were harvested with Accutase^®^ solution (Sigma-Aldrich, St. Louis, MO, USA) and were subjected to an analysis of the cell cycle distribution or an analysis of the cell death mechanism (apoptosis vs. necrosis and overall death rate).

#### 4.10.1. Analysis of the Cell Death Mechanism

Apoptosis and necrosis were assessed by staining with propidium iodide (PI, Thermo Fisher Scientific, Waltham, MA, USA) and AnnexinV-FITC (BioLegend™, San Diego, CA, USA) according to manufacturers’ specifications. Cell samples were then analyzed with a BD FACS Canto II flow cytometer (Becton Dickinson, Francklin Lakes, NJ, USA) and data were analyzed via the cytobank platform (https://www.cytobank.org/, accessed date: January 2019–June 2021, Cytobank, Inc., Santa Clara, CA, USA). Cells without staining were considered alive/vital, while cells which were single-stained by PI necrotic, those single-stained by AnnexinV-FITC apoptotic and those which were stained by PI and AnnexinV-FITC were considered dead.

#### 4.10.2. Analysis of the Cell Cycle Distribution

For cell cycle analysis, cells were harvested, washed with PBS, and finally suspended in 500 μL PBS and 4.5 mL 70% ice-cold ethanol. After incubation for at least 2 h at −20 °C, cells were washed with washing buffer (0.2% (*v/v*) Triton X-100, 1% BSA in PBS) and transferred to propidium iodide staining solution (0.1% (*w/v*) RNAse A, 5 μg/mL PI in PBS). The staining was evaluated with BD FACS Canto II flow cytometer (Becton Dickinson, Francklin Lakes, NJ, USA) and analyzed with the ModFit LT software (Verity Software House, Topsham, ME, USA).

### 4.11. Expression Analysis by SDS-PAGE and Western Blotting

For expression analyses, cells were harvested and lysed after the indicated treatment. In this process, 30–40 μg of total protein was loaded onto 12% acrylamide gels and were subjected to SDS-PAGE. Gels were transferred to Immobilon^®^-P PVDF Membrane (Merck Millipore Ltd., Billerica, MA, USA) by means of the wet Western blotting procedure. We used the following antibodies: p21 Waf1/Cip1 (12D1) rabbit mAB (#2947 Cell Signaling Technology^®^, Danvers, MA, USA), CDKN2A/p16INK4a rabbit polyclonal antibody (#ab189302, Abcam, Cambridge, UK) ß-Actin mouse mAB (A5441, Sigma-Aldrich, St. Louis, MO, USA), GAPDH mouse AB (# 9484 Abcam, Milton, UK), P53 mouse mAB clone DO-1 (#P6874 Sigma-Aldrich, St. Louis, Missouri, USA), anti-mouse IgG, HRP-linked Antibody (#7076 Cell Signaling Technology^®^, Danvers, MA, USA), anti-rabbit IgG, HRP-linked Antibody (#7074 Cell Signaling Technology^®^, Danvers, MA, USA). Blots were developed by Western Lightning Plus ECL (Perkin Elmer, Waltham, MA, USA), documented with the ChemiDoc Imager (Bio-Rad, Hercules, CA, USA), and evaluated with the Image Lab software (version 5.0 build 18, Bio-Rad). The measured intensities were normalized by the housekeeping gene ß-actin, each experiment was repeated at least three times, and the relative expression was compared between the experimental groups on one blot. The relative expression is expressed as percent of the corresponding control.

### 4.12. Statistical Analysis

One-way and two-way ANOVA and unpaired two-tailed *t*-tests followed by Bonferroni correction for multiple comparisons and Welch correction for uneven variations, where applicable, were used to assess statistical significances. All calculations were performed using the software GraphPad Prism 6 for Windows, Version 6.01 (GraphPad Software, La Jolla, CA, USA). The data shown are means ± SD, unless otherwise stated, * *p* < 0.05, ** *p* < 0.01, *** *p* < 0.001.

## 5. Conclusions

In this study, we found three main points. First, we showed that 16 Gy gamma-irradiation induced a senescent state in the irradiated cancer cell lines. Second, we showed that after only a few weeks, the tumor cells were able to escape that senescent state and proliferate again. Finally, we showed that ZnO NPs were capable of eliminating residual senescent tumor cells. This reveals the potential of ZnO NPs as an innovative adjuvant tumor therapy that could break through radioresistance associated with radiation-induced senescence.

## Figures and Tables

**Figure 1 cancers-13-02989-f001:**
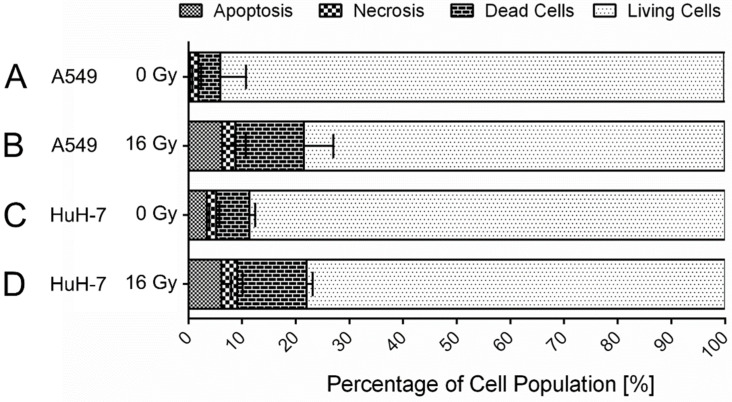
Cell death after 16 Gy gamma-irradiation. Irradiation resulted in a reduction in the viable cell population by ten to fifteen percent in A549 (*p* = 0.03) (**A**,**B**) and in HuH-7 (*p* = 0.05) (**C**,**D**) cells within 72 h. The percentage of living cells in the cell population was decreased to approximately 80% in both investigated cell lines. Shown are mean values ± SD, unpaired, two-tailed t-test, comparison between untreated control cells and irradiated cells of the respective cell line, N ≥ 3.

**Figure 2 cancers-13-02989-f002:**
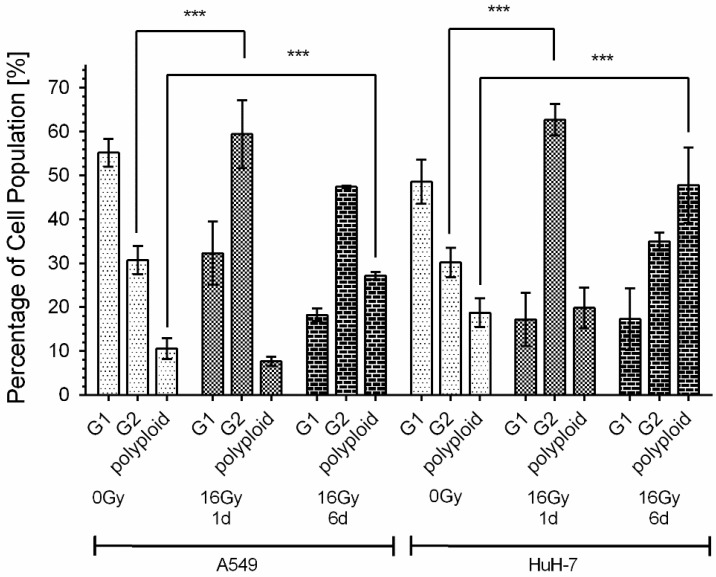
Cell cycle arrest and increase in the polyploid cell population after irradiation. HuH-7 and A549 cells showed a significant decrease in the percentage of cells in the G1 phase and an increase in the percentage of cells in the G2 phase within 24 h after 16 Gy gamma-irradiation. On day six after irradiation, a percentual increase in polyploid tumor cells was observed in both cell lines. Shown are mean values ± SD, two-way ANOVA, comparison between untreated control cells and irradiated cells of the respective cell line on day one (1 d) and day six (6 d) after irradiation, correction for multiple comparisons by Bonferroni, N ≥ 4, *** *p* < 0.001.

**Figure 3 cancers-13-02989-f003:**
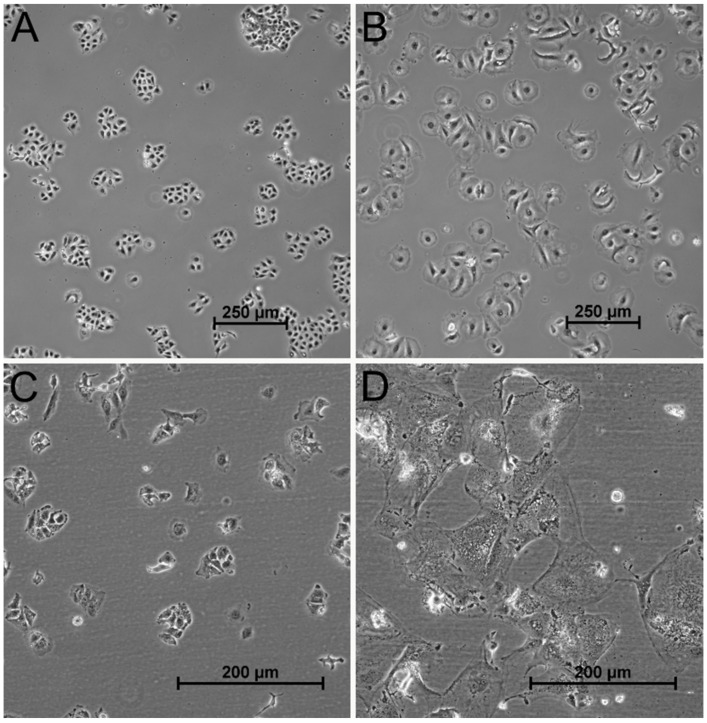
Cell morphology after 16 Gy gamma-irradiation. A549 (**A**,**B**) and HuH-7 (**C**,**D**) cells showed a pronounced increase in cell size after 16 Gy gamma-irradiation. Shown are representative images of untreated A549 cells (**A**) and HuH-7 cells (**C**) compared to cells six days after treatment with 16 Gy irradiation (**B**: A549; **D**: HuH-7). Cells of both tumor cell lines were swollen after irradiation and showed the typical “fried egg”–morphology.

**Figure 4 cancers-13-02989-f004:**
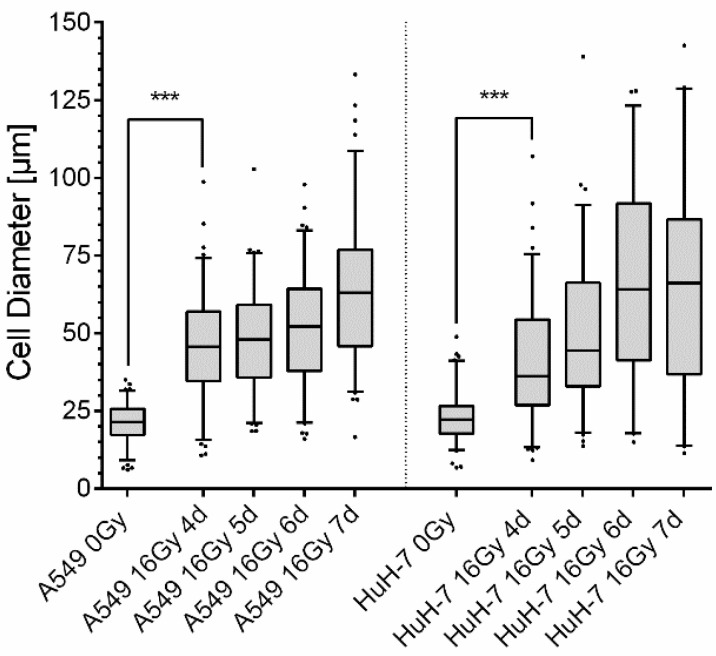
Increase in cell diameter after 16 Gy gamma-irradiation. After irradiation, the diameter of A549 cells (left) and HuH-7 cells (right) increased significantly until day four (4 d), followed by a slower but steady increase in cell diameter until day seven (7 d) after irradiation. Within the first four days, the cell diameter increased sharply from 20.9 µm ± 6.5 µm to 45.6 µm ± 17.6 µm in A549 cells and from 23.2 µm ± 7.9 µm to 40.5 µm ± 19.4 µm in HuH-7 cells. In the following days, the increase in cell diameter leveled off and reached values of 61.1 µm ± 23.2 µm in A549 cells and 66.3 µm ± 32.4 µm in HuH-7 cells on the seventh day after irradiation. Concomitantly, the size distribution of the cell population broadened. Shown are box blots with 5–95% percentiles and median, outliers are shown as dots, unpaired two-tailed *t*-test, comparison between untreated control cells and irradiated cells at the different time points, *** *p* < 0.001.

**Figure 5 cancers-13-02989-f005:**
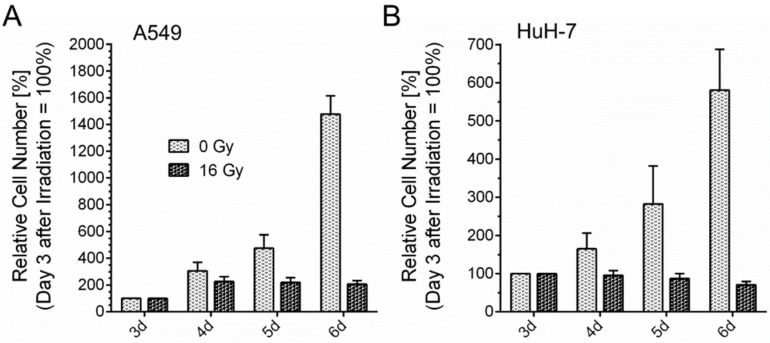
Stopping of proliferation after 16 Gy gamma-irradiation. Proliferation of A549 (**A**) and HuH-7 (**B**) cells was stopped on day four after irradiation. Cell numbers are shown as relative cell numbers with respect to day three (3 d = 100%) after irradiation and compared with normal cell proliferation of untreated control cells of each cell line. Mean values + SD are shown.

**Figure 6 cancers-13-02989-f006:**
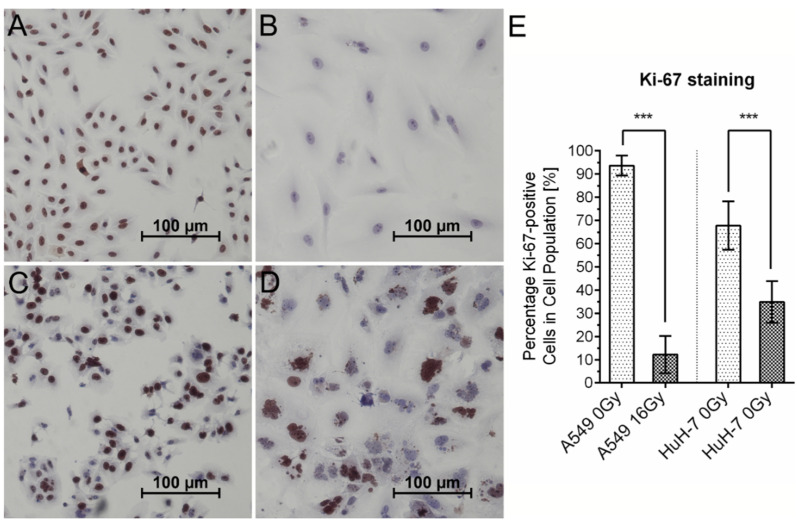
Decrease in Ki-67expression after irradiation. Representative images of A549 (**A**,**B**) and HuH-7 (**C**,**D**) cells stained immunohistochemically for Ki-67. Comparison of untreated control samples (**A**,**C**) and irradiated cells (**B**,**D**) revealed significantly decreased Ki-67 expression after irradiation (**E**). Mean values ± SD are shown, unpaired two-tailed t-test, comparison between untreated control cells and irradiated cells, *** *p* < 0.001.

**Figure 7 cancers-13-02989-f007:**
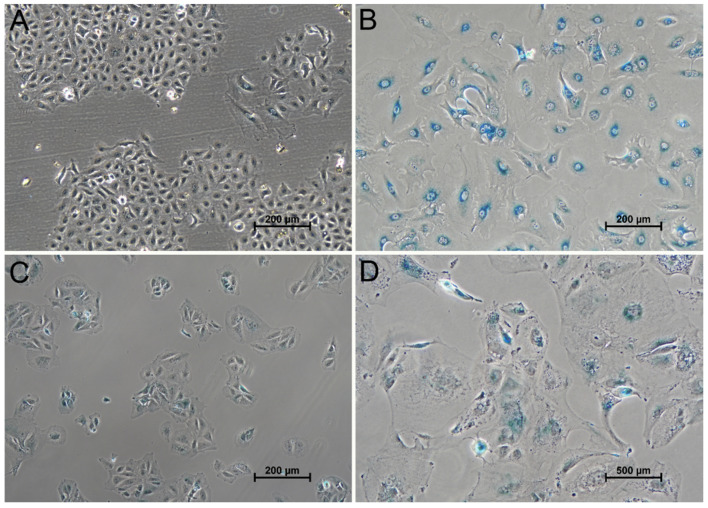
Senescence-associated β-galactosidase staining of A549 and HuH-7 cells. X-Gal-staining of A549 (**A**,**B**) and HuH-7 (**C**,**D**) revealed mostly negative cells in untreated control samples (**A**,**C**). Isolated cells showed weak X-Gal positive staining, which was accompanied by a larger cell size and a slightly different cell morphology compared to those cells that were X-Gal-negative. After irradiation with 16 Gy, almost all A549 cells became X-Gal-positive (99%) and most HuH-7 showed positive staining (81%).

**Figure 8 cancers-13-02989-f008:**
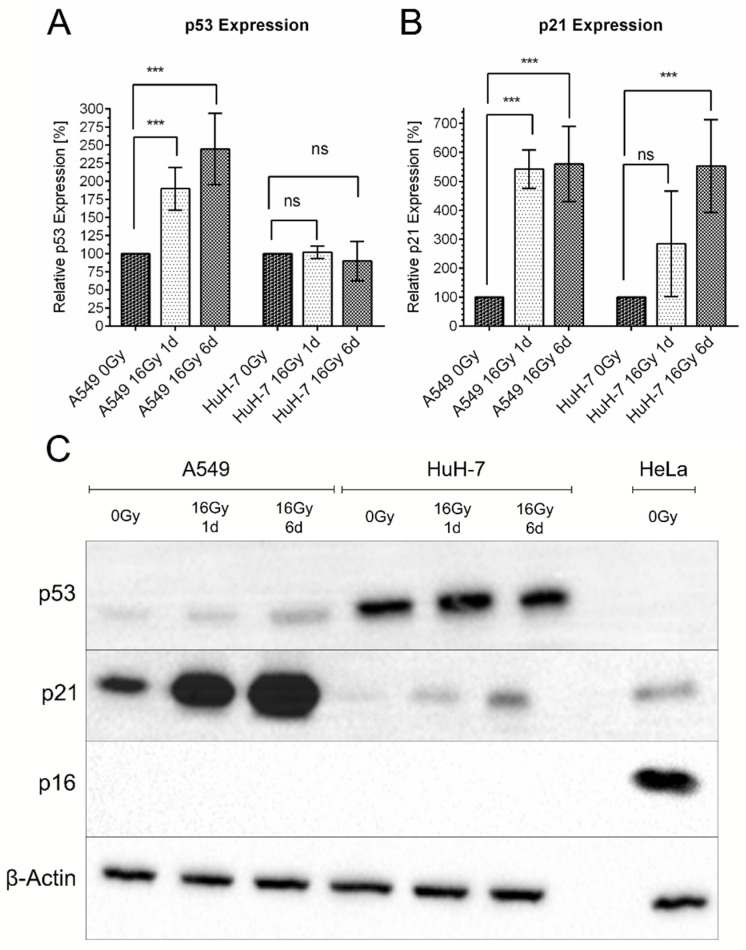
Expression levels of p16, p53, and p21 in A549 and HuH-7 cells after irradiation. The relative expression of p53 increased significantly in A549 cells after irradiation. In contrast, in HuH-7 cells, p53 expression remained at a consistently very high level (**A**). Following irradiation, the expression of p21 increased in both cell lines under investigation, in A549 cells faster (already on day one after irradiation) than in HuH-7 cells (**B**). There was no detectable p16 expression in A549 and HuH-7 cells at all time points and under all treatments under investigation. In (**C**), a representative Western blot is shown, showing expression of p53, p21, p16, and β-actin. The latter served as non-regulated reference protein. HeLa cells served as p16 positive control. Shown are mean values ± SD, the relative expression of each protein studied is expressed with respect to the non-regulated reference protein β-actin and related to the non-irradiated control sample (=100%) of the corresponding cell line. Two-way ANOVA, comparison between untreated control cells and irradiated cells at day one (1 d) and day 6 (6 d) of the respective cell line, correction for multiple comparisons by Bonferroni, N ≥ 4, *** *p* < 0.001, ns = not significant.

**Figure 9 cancers-13-02989-f009:**
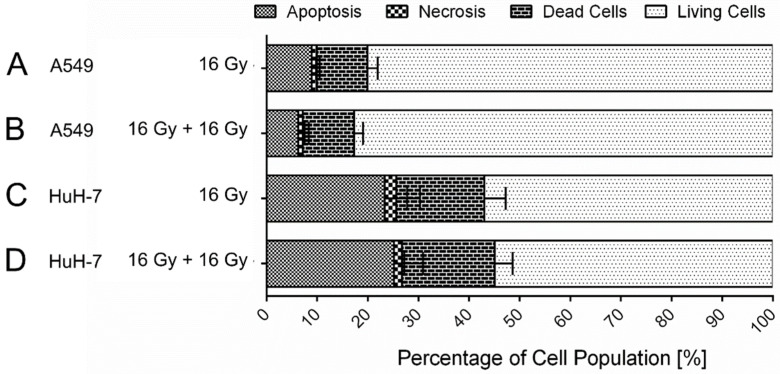
Percentage of apoptotic, necrotic, dead, and living cells of A549 (**A**,**B**) and HuH-7 (**C**,**D**) cells after one (**A**,**C**) and two (**B**,**D**) treatments with 16 Gy gamma-irradiation each. A second 16 Gy-treatment six days after the first irradiation treatment did not increase the dying cell (apoptotic, necrotic, and dead cells) fraction significantly in both studied cell lines. Shown are mean values ± SD, unpaired, two-tailed t-test, comparison between cells treated with 1 × 16 Gy and 2 × 16 Gy of the respective cell line, N ≥ 3.

**Figure 10 cancers-13-02989-f010:**
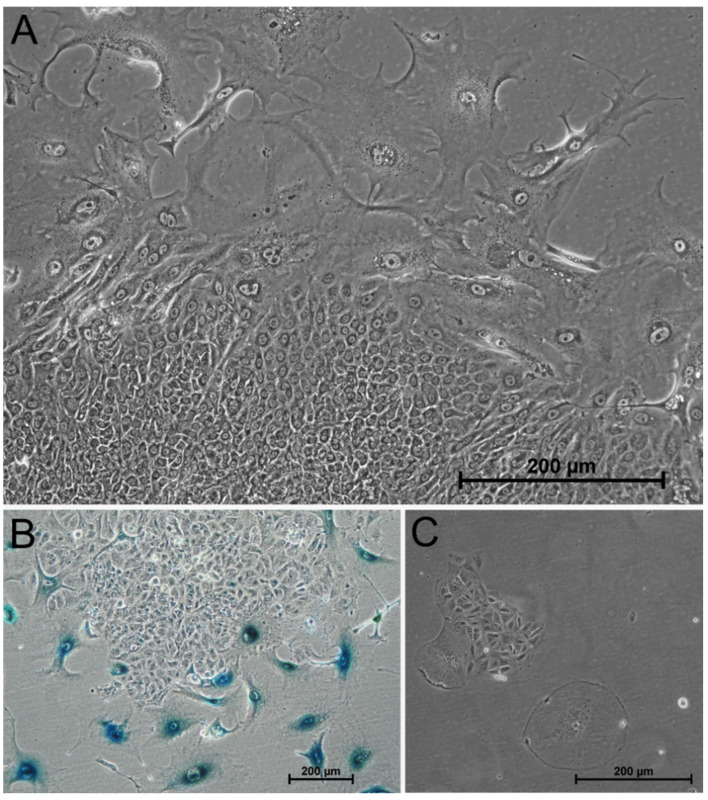
Regrowing tumor cells of the cell lines A549 and HuH-7 weeks after treatment with 16 Gy. Section (**A**) shows the morphology of A549 cells 70 days after treatment with 16 Gy. Small, highly proliferative cells appeared between the enlarged senescent cells and formed islands that began to grow and occupy progressively more space within the cell population. Their morphology indicated an escape from the senescent state. This was confirmed by senescence-associated β-galactosidase staining (**B**) of A549 cells 52 days after treatment with 16 Gy. It showed that the small cells in the islands lost positive staining. Additionally, HuH-7 cells were able to regrow after 28 days after 16 Gy irradiation (morphology shown in **C**). Between the giant senescent HuH-7 cells, small islands of proliferating tumor cells appeared again.

**Figure 11 cancers-13-02989-f011:**
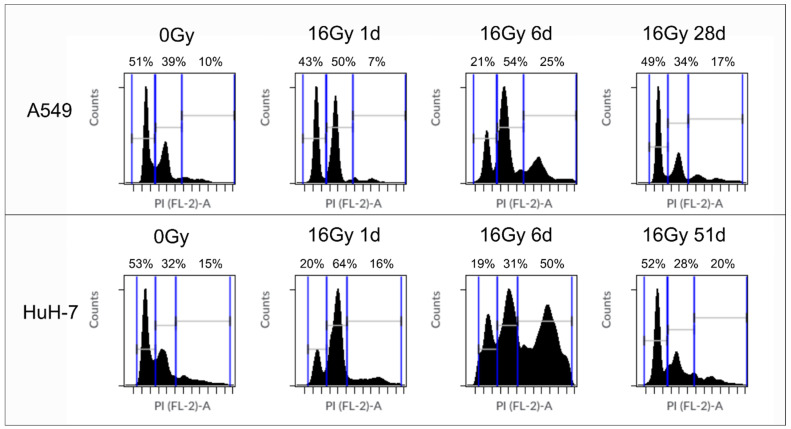
Cell cycle distribution of A549 and HuH-7 cells before irradiation, immediately after 16 Gy gamma-irradiation and after the appearance of a new population of non-senescent cells after 28 days (A549) and 51 days (HuH-7) after irradiation. The majority of the untreated A549 and HuH-7 cells was in the G1 phase. Between day one and six after irradiation with 16 Gy both cell lines entered G2 cell cycle arrest. After 28 or 51 days, respectively, the cell cycle distribution reversed, and most of the cells were in the G1 phase again. Two representative cell cycle measurements are shown. Above each figure, we noted the percentage of cells in the corresponding cell cycle phase: the first number indicates the percentage of cells in the G1 phase, the second the percentage of cells in the G2 phase, and the last number indicates the percentage of polyploid cells.

**Figure 12 cancers-13-02989-f012:**
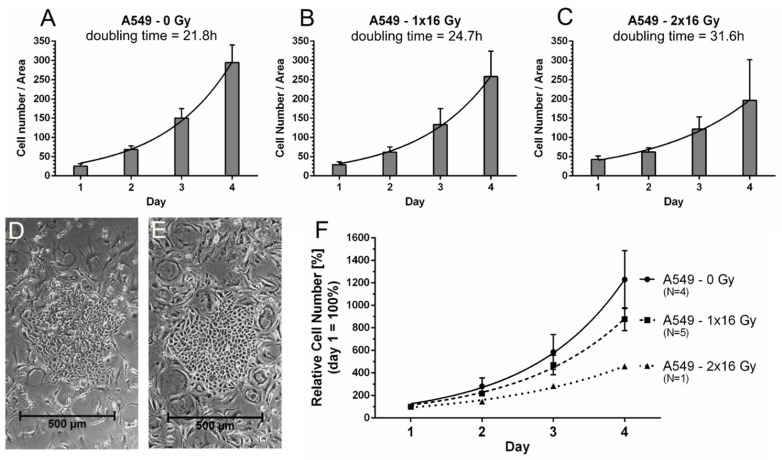
Proliferation and doubling times of A549 cells before irradiation and after regrowth of colonies of non-senescent cells several weeks after 1 × 16 Gy and 2 × 16 Gy gamma-irradiation. After counting the cell number per randomly chosen area on four consecutive days, an exponential growth equation was used to extrapolate the doubling time of the respective tumor cell population. The doubling time of untreated A549 cells was calculated to be 21.8 h on average (N = 4, R^2^ = 0.91). After 1 × 16 Gy gamma-irradiation and the regrowth of proliferating tumor cells out of the initially senescent cell population, a doubling time of 24.7 h on average (N = 5, R^2^ = 0.96) was seen, and after 2 × 16 Gy, the doubling time slowed down to 31.6 h on average (N = 1, R^2^ = 0.99). Exemplary growth curves for each group are shown in **A** (0 Gy), **B** (1 × 16 Gy), and **C** (2 × 16 Gy). A representative proliferating tumor cell colony at day 15 after 1 × 16 Gy is shown in **D**. The partial image **E** shows the regrowth of proliferating tumor cells 27 days after 2 × 16 Gy. The second irradiation cycle was applied six days after the first one. From the single cell counts, general growth curves for untreated A549 cells, and those which survived 1 × 16 Gy, or 2 × 16 Gy were extrapolated (**F**). It can be seen that proliferative capacity is regained after gamma-irradiation, but cell doubling slows down.

**Figure 13 cancers-13-02989-f013:**
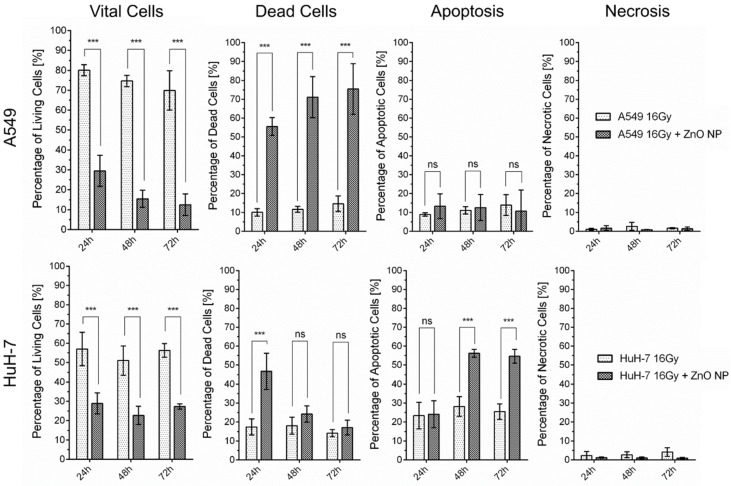
Treatment of senescent A549 and HuH-7 cells with zinc oxide nanoparticles increased the percentage of dead cells significantly and decreased the percentage of vital cells significantly within 24 h as measured by flow cytometry. Shown are mean values ± SD. Two-way ANOVA, comparison of senescent cells and senescent cells treated with 100 µg/mL ZnO NPs of the respective cell line at the indicated time-point post-treatment. Correction for multiple comparisons by Bonferroni, N ≥ 3, *** *p* < 0.001, ns = not significant.

## Data Availability

The data presented in this study are available on request from the corresponding author. The data are not publicly available due to the policy of research projects.

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
