# Peer review of "Zinc Oxide Nanoparticles Can Intervene in Radiation-Induced Senescence and Eradicate Residual Tumor Cells"

_cancers, 2021, doi:10.3390/cancers13122989_

Round 1
Reviewer 1 Report
The study by Wiesmann et al is a a very interesting and well written paper, contributing to the clarification on the role of radiation induced senescence in the outcome of cancer radiotherapy. A major comment, however, concerns the very interesting observation that irradiation of cells with a second fraction of 16Gy did not increase cell killing.
- This is an important finding that should be further illustrated and elaborated. At what time point did the 2nd irradiation happen, This should be clearly stated in the text (not only in the figure). How and when the cell proliferation was measured compared cells receiving only one RT. As stated in pragraph 2.3, the heavily irradiated cells with 16Gy started regrowing 3-4 weeks after irradiation. In this way comparison on the effect of the 2nd 16Gy fraction on cell proliferation ability should refer to proliferation rates after the 3-4th week. Delays on the onset of regrowth between the two dose groups should also be mentioned.
- What was the proliferation rate of regrowing senescent cells compared to the non-irradiated cells? What was the proliferation rate of regrowing senescent cells after the 2nd irradiation compared to the non-irradiated cells and only once irradiated cells?
Author Response
Response to Reviewer 1 Comments
Point 1: […] A major comment, however, concerns the very interesting observation that irradiation of cells with a second fraction of 16Gy did not increase cell killing. This is an important finding that should be further illustrated and elaborated. […]
Answer: Thank you very much for drawing our attention to this indeed very interesting aspect of our study. Inspired by your comments, we went back to the lab and examined the proliferation of the cells before and after treatment with 1x16 Gy and 2x16 Gy again in detail. The newly gained insights have been incorporated into Fig. 12 and are described in lines 287-296. In short, we could observe, that A549 tumor cells regain their proliferative capacity to a great extent after regrowth out of the senescent cell populations.
Point 2: […] At what time point did the 2nd irradiation happen? This should be clearly stated in the text (not only in the figure). […]
Answer: The second irradiation was applied six days after the first one. We added that information now also in the text in the results section (lines 232-233) and in the materials and methods section (line 489-491) to clarify that point.
Point 3: […] How and when the cell proliferation was measured compared to cells receiving only one RT. […]
Answer: First, cell proliferation was measured in the original tumor cell population prior to treatment. Immediately after irradiation, the tumor cells entered senescence, and proliferation was stopped. This is shown in Fig. 5 (line 175). Cell proliferation started again in colonies of non-senescent tumor cells which appeared within two to four weeks after irradiation treatment. To measure the doubling times and growth rates of these tumor cells, colonies were picked when they were visible with the naked eye (approximately 50-200 cells per colony), seeded in a new culture plate and cell numbers in randomly chosen areas were counted on four consecutive days. Cell numbers were then used to extrapolate an exponential growth curve and thus calculate the respective doubling times. The results of these experiments are described in lines 287-296, they are depicted in Fig. 12 and the method to measure proliferation is described in detail in lines 538-550.
Point 4: […] As stated in paragraph 2.3, the heavily irradiated cells with 16Gy started regrowing 3-4 weeks after irradiation. In this way comparison on the effect of the 2nd 16Gy fraction on cell proliferation ability should refer to proliferation rates after the 3-4th week. […]
Answer: Thank you very much for that valuable input. That is exactly what we have now done in further experiments inspired by your comments. We measured proliferation of tumor cells, before treatment, after regrowth from senescent cell populations after 1x16 Gy and also after regrowth from senescent cell populations after two cycles of 16 Gy irradiation (lines 287-296, Fig. 12).
Point 5: […] Delays on the onset of regrowth between the two dose groups should also be mentioned. […]
Answer: We found that regrowth of tumor cells after two cycles of 16 Gy irradiation was slightly delayed compared to regrowth after 1x16 Gy and we also added that information to the manuscript (line 297-300). The exact time point of regrowth was individual for every cell culture plate, that was irradiated. Thus, we cannot give an exact time point at which tumor cells will start to proliferate again. Sometimes as soon as nine days after irradiation the first proliferating tumor cell colonies appeared after 1x16 Gy, sometimes it took longer. In most cultures proliferating tumor cells appeared in the second or third week after irradiation, if any appeared. However, sometimes also as long as 50 or 60 days after irradiation islands of proliferating tumor cells were spotted. After 2x16 Gy colonies were visible 22 days after the first irradiation cycle (16 days after the second irradiation) at the earliest.
Point 6: […] What was the proliferation rate of regrowing senescent cells compared to the non-irradiated cells? What was the proliferation rate of regrowing senescent cells after the 2nd irradiation compared to the non-irradiated cells and only once irradiated cells?
Answer: We found that the cells that were regrowing out of the formerly senescent tumor cells nearly reached the proliferative potential they had before irradiation. The doubling time of A549 tumor cells before 16 Gy gamma-irradiation was 21.8 h. After the survival of 1x16 Gray gamma-irradiation and regrowth, the doubling time of the regrown cells was 24.7 h. Thus, multiplication slightly slowed down. The doubling time of A549 cells after 2x16 Gy was 31.6 h, thus the multiplication potential of these cells was decreased in comparison to the original cell population. These findings are depicted in lines 287-296 in the results section and discussed in lines 434-441.

Reviewer 2 Report
This study shows that two cell lines (A549 and HuH-7) enter a senescence state after irradiation with 16Gy of gamma-irradiation. The authors also show that these senescent cells are sensitive to Zinc Oxide Nanoparticles but not to a new dose of gamma-irradiation.
The study is well designed and easy to read. I find it interesting and original, and therefore I recommend it for publication. However, I would like to see some original or referenced data on the toxicity of these NPs in healthy cell lines. I couldn't find this data either in this paper or in reference 22.
It is hard to evaluate the interest of these synthesized particles without knowing if they can specifically kill cancer cells of all cells in general.
Author Response
Response to Reviewer 2 Comments
Point 1: […] The study is well designed and easy to read. I find it interesting and original, and therefore I recommend it for publication. However, I would like to see some original or referenced data on the toxicity of these NPs in healthy cell lines. I couldn't find this data either in this paper or in reference 22. It is hard to evaluate the interest of these synthesized particles without knowing if they can specifically kill cancer cells of all cells in general.
Answer: Thank you very much for that comment. We totally agree with you that it is important to know about the toxicity of ZnO NPs in normal cells in order to be able to evaluate their relevance for tumor therapy. This information can indeed be found in reference 22. There we describe that certain ZnO NP concentrations show higher toxicity towards tumor cells than towards healthy fibroblasts. Furthermore, there is the possibility to cover ZnO NPs with a silica shell which allows for active targeting with targeting moieties to ensure increased accumulation of the NPs in tumor tissue, where they disintegrate and exert their toxicity. These aspects were also added to the manuscript (lines 442-447) in order to clarify the interest of the particles for tumor treatment.

Round 2
Reviewer 1 Report
Authors have adequately responded to my comments.